# Aliens on the Road: Surveying Wildlife Roadkill to Assess the Risk of Biological Invasion

**DOI:** 10.3390/biology12060850

**Published:** 2023-06-13

**Authors:** Andrea Viviano, Marcello D’Amico, Emiliano Mori

**Affiliations:** 1Istituto di Ricerca sugli Ecosistemi Terrestri, Consiglio Nazionale delle Ricerche, 50019 Sesto Fiorentino, Italy; 2Department of Conservation Biology and Global Change, Doñana Biological Station, Spanish National Research Council (CSIC), 41092 Seville, Spain; 3National Biodiversity Future Center, 90133 Palermo, Italy

**Keywords:** alien species, allochthonous species, biological invasions, introduced species, invasive species, non-native species, road ecology, roadkill, vertebrates, wildlife–vehicle collisions

## Abstract

**Simple Summary:**

Monitoring non-native species is important to assess their invasion risk. Here, we carried out a literature review on roadkill studies to investigate geographical patterns of biological invasions. We believe that roadkill data from published literature can be a valuable resource for people dealing with biological invasions, especially when field surveys cannot be performed. We retrieved 2314 studies published until January 2022, but only 40 of them (in addition to our own unpublished field data) included a full list of road-killed species (with a number of affected individuals for each species), so we were able to include them in our analyses. We classified all road-killed species from retrieved studies as native or introduced (domestic, introduced in historical times or recently released). We found that a higher number of introduced species would be recorded among roadkill in Mediterranean and Temperate regions with respect to Tropical and Desert areas. This result is in line with the current knowledge on non-native species distribution at the global scale, thus confirming that roadkill datasets can be used beyond the study of road impacts, such as for an assessment of different levels of biological invasions among different countries.

**Abstract:**

Monitoring the presence and distribution of alien species is pivotal to assessing the risk of biological invasion. In our study, we carried out a worldwide review of roadkill data to investigate geographical patterns of biological invasions. We hypothesise that roadkill data from published literature can turn out to be a valuable resource for researchers and wildlife managers, especially when more focused surveys cannot be performed. We retrieved a total of 2314 works published until January 2022. Among those, only 41 (including our original data) fitted our requirements (i.e., including a total list of roadkilled terrestrial vertebrates, with a number of affected individuals for each species) and were included in our analysis. All roadkilled species from retrieved studies were classified as native or introduced (domestic, paleo-introduced, or recently released). We found that a higher number of introduced species would be recorded among roadkill in Mediterranean and Temperate areas with respect to Tropical and Desert biomes. This is definitely in line with the current knowledge on alien species distribution at the global scale, thus confirming that roadkill datasets can be used beyond the study of road impacts, such as for an assessment of different levels of biological invasions among different countries.

## 1. Introduction

In recent decades, the percentage of the planet covered by roads has increased worldwide [1,2,3], together with their impacts on biodiversity and ecosystems [4,5,6]. Fragmentation, light, noise and chemical pollution, as well as direct human disturbance, are among the most well-recognised road impacts [7,8,9]. Other impacts have also been described, such as roads promoting biological invasions, especially for plants [10,11,12], but also for some animal species [13,14,15]. However, wildlife-vehicle collisions are the most evident road impact, as well as the most investigated one in scientific literature [16,17,18].

Roadkill is so well studied because it can affect the persistence of animal populations [18,19,20], mainly by limiting dispersal possibilities and then disrupting metapopulations, thus triggering local extinction vortices [21,22,23]. Such extinctions are often a relevant issue also for economic reasons because, in some cases, they concern species providing valuable ecosystem services [24,25,26]. Furthermore, roadkill is so well studied because they represent a major and often overlooked issue for road safety, especially when involving large vertebrates [27,28,29].

Roadkill observations and even surveys are increasingly used to monitor and update species distribution, e.g., [30,31,32,33,34], by means of citizen science, e.g., [35,36,37]. Individual hobbyists and wildlife enthusiasts have greatly increased in the last 30 years, and the majority of them upload opportunistic records to online platforms [38,39,40], including projects on road mortality [35,41,42]. Similarly, the low cost of detecting and uploading roadkill to online databanks (e.g., iNaturalist, GBIF) has also been used to monitor biological invasions, particularly at early invasion stages [43,44,45]. Immediately after release or escape, invasive alien species are present in new areas at low densities, which may prevent their detection through standardised monitoring, therefore relying on accidental observations, such as roadkill as first detection [46,47,48]. For instance, in Central Italy, the monitoring of the invasive population of the Northern raccoon *Procyon lotor* started after the first detections of several roadkilled individuals [49]. Furthermore, given that often roadkilled individuals are juveniles in dispersal [50,51,52], the detection of invasive alien species may also provide evidence of the potential range of expansion of established populations [53,54,55].

The effectiveness of detecting invasive alien species or monitoring their distribution through roadkill has never been quantified, although field evidence of the successfulness of this method has been suggested [34,56,57]. Using roadkill to study alien species represents a very novel approach to biological invasions, and to the best of our knowledge, there are no studies on roadkill of alien species comparing different locations. With this work, we aimed to fill this gap, and we hypothesised that the percentage of roadkilled allochthonous species varies among regions, being higher in the Mediterranean biome (first prediction), especially in Italy (second prediction). In fact, the Mediterranean biome is one of the world’s biodiversity hotspots, but it is also threatened by a high number of alien species [58,59,60], probably because it includes, at the same time, the highest biodiversity but also the highest levels of human pressure and trade [61,62,63]. In the Mediterranean biome, Italy is reported to be one of the richest countries in animal diversity [62,64,65], as well as one of the richest in alien species (nearly 17% of all mammals and over 57% of freshwater fish species are alien in Italy [66,67]). This is probably because of the intense trade pressure which has occurred in this area for millennia [68,69,70].

In this work, we carried out a roadkill survey focused on terrestrial vertebrates in our study area in a Mediterranean region of Central Italy (i.e., Tuscany), and then we compared our list of alien vs. native roadkilled species with all published and comparable lists from other regions of the world (including both Mediterranean and other biomes), which we obtained by a literature review. Into the framework of such comparisons, we tested if the percentage of roadkilled introduced species varied among biomes (and then countries), always considering several possible confounding factors at the same time, such as the latitude, Gross Domestic Product (GDP hereafter), and the number of neighbouring countries. High temperatures extend the available time for growth and reproduction, so low latitudes can comparatively host higher numbers of introduced species [71,72,73]. GDP has also been described as increasing the number of introduced species, probably because of the combined effects of trade, consumerism and urbanization [74,75,76]. Finally, the number of introduced species is usually higher in connected countries with established trade and tourism along shared infrastructures, such as neighbouring countries [74,77,78]. In order to investigate the reliability of roadkill data for investigating biological invasions at a macro-ecological scale, we also carried out an additional analysis comparing the whole lists of alien vs. native species inhabiting all the countries considered in the first analysis, always including the same approach and candidate predictors. This study represents a first example of how we can investigate patterns of biological invasions by analysing roadkill datasets, which can become a valuable resource for researchers and wildlife managers, especially for first assessments when more focused surveys cannot be performed.

## 2. Materials and Methods

### 2.1. Study Area (Fieldwork Survey)

Our fieldwork was conducted in Southern Tuscany (Provinces of Grosseto and Siena, Central Italy). We travelled once a week along the road linking the village of Prata (Province of Grosseto) to Grosseto and Siena for three years (2011–2013), for a total of about 101 km/week (Figure 1).

The road skirts a high number of habitat types, including dense deciduous and mixed woodlands (about 27%), farmlands (about 43%), scrublands with hedgerows (16%), human settlements (9%) and fallows (5%). The whole road was paved, and an allowed speed limit was 70–90 km/h throughout most road sections, apart from 16 km (in total) of the beltway (speed limit: 110 km/h) in the surroundings of the main cities (Siena and Grosseto). The study area is characterised by the presence of at least 38 mammal species (represented among others by the wild boar *Sus scrofa*, the roe deer *Capreolus capreolus*, the red fox *Vulpes vulpes*, the European badger *Meles meles* and the crested porcupine *Hystrix cristata*), about 100 bird species (also including the common pheasant *Phasianus colchicus*, the red-legged partridge *Alectoris rufa* and over 15 raptor species), 16 species of reptiles (e.g., the asp viper *Vipera aspis* and the Italian wall lizard *Podarcis siculus*) and eight species of Amphibians (e.g., the common toad *Bufo bufo* and the Italian crested newt *Triturus carnifex*) [79,80,81]. The local community of terrestrial vertebrates mainly consists of common species, but some threatened species are also present, such as the European turtle-dove *Streptopelia turtur* (globally listed as Vulnerable; [82]).

### 2.2. Data Collection and Analyses (Fieldwork Survey and Literature Review)

During our fieldwork survey, every time a roadkilled animal was detected, it was georeferenced and classified (when possible) at the species level. We also recorded whether the species was native or introduced (see Appendix A). Using the list of roadkilled species resulting from our fieldwork survey, we calculated the percentage of introduced species on the total of roadkilled species in order to compare this percentage to the same value from all published and comparable lists from other regions of the world. Regarding introduced species, we also specified the percentage of recently introduced, paleo-introduced and domestic species (we considered all the species introduced before 1500 AD as paleo-introduced [83,84]).

We performed our first literature review by searching those lists of roadkilled terrestrial vertebrates on the Web of Science (WOS; [85]) by implementing the following search string: (“roadkill*” OR “road-kill*” OR (“collision” AND “road”) AND (“vertebrate*” OR “wildlife” OR “animal*”). We retained articles focusing on the whole community of terrestrial vertebrates and including an explicit list of roadkilled species. Once we obtained the final list of retained articles, we extracted the study areas in order to perform a second literature review aimed at obtaining the percentage of introduced species on the total of species inhabiting those countries (always focusing on terrestrial vertebrates). We completed this second review also with information proceeding from institutional websites such as www.iucngisd.org, www.iucnredlist.org, www.griis.org (all accessed on 12 February 2023), and several national websites providing inventories of introduced and native species.

For each study area mentioned in those articles, we extracted the country, the biome (as defined by [86,87]), and three explanatory variables potentially affecting the number of introduced species in a given area, such as latitude, GDP, and the number of neighbouring countries, in addition to the aforementioned percentage of roadkilled introduced species. Considering that the analysis would probably have relatively little data, we modified some of the explanatory variables in order to have fewer levels within each of them. More concretely, we combined some biomes (i.e., “Temperate grasslands, savannas, and shrublands” and “Temperate broadleaf and mixed forests” in “Temperate biomes”, “Tropical and subtropical grasslands, savannas, and shrublands” and “Tropical and subtropical moist broadleaf forests” in “Tropical biomes”), we implemented the latitude as a continuous variable without decimals, we transformed the GDP in an ordinal variable with four levels (i.e., less than USD one billion, between USD one and two billion, between USD two and three billion, and more than USD three billion), and we also transformed the number of neighbouring countries into an ordinal variable with four levels (i.e., without neighbouring countries, with one/two neighbouring countries, or with many neighbouring countries).

In order to test our first prediction (i.e., more introduced species in Mediterranean biomes), we used generalised linear mixed models (GLMMs; negative binomial error distribution and log link function) and evaluated their performance using Akaike Information Criterion (AIC). For each model, the response variable was the percentage of introduced species on the total of roadkilled species, considering the country of each retained study as a random factor. We compared all the possible univariate models, including the considered explanatory variables, with the full model and also the null model (i.e., only including the random factor). We selected the most supported model using AICc (i.e., AIC for small sample sizes) and calculated Akaike weights (wAICc) to estimate the relative support for every model (ranging from 0 to 1, with larger numbers indicating greater support; [88,89,90]). We also provided R^2^ for each considered model. Additionally, we repeated exactly the same AIC model selection of the same GLMMs, but this time implementing as a response variable the percentage of introduced species on the total of existing species.

In order to test our second prediction (i.e., more introduced species in Italy than in other countries of Mediterranean biomes), we performed a generalised linear model (GLM) in which the response variable was again the percentage of introduced species on the total of roadkilled species, and the explanatory variable was the country of each retained study. We again used negative binomial error distribution and log link function. We evaluated model performance by comparing the AIC of the considered model with the AIC of the related null model.

## 3. Results

### 3.1. Fieldwork Survey

In Central Italy, we collected a total of 511 roadkill specimens belonging to 89 species (Figure 1; Appendix A). Most roadkill belonged to Mammals (N = 273, for a total of 22 species), followed by Birds (N = 137, for a total of 53 species), Reptiles (N = 53, for a total of ten species) and Amphibians (N = 48, for a total of four species). Introduced species included two domestic (domestic cat *Felis catus* and domestic pigeon *Columba livia domestica*), four paleo-introduced (fallow deer *Dama dama*, crested porcupine *Hystrix cristata*, black rat *Rattus rattus*, and common pheasant *Phasianus colchicus*) and two recently introduced species (Iberian pond turtle *Mauremys leprosa* and coypu *Myocastor coypus*). Two further species (i.e., red-legged partridge *Alectoris rufa* and grey partridge *Perdix perdix*), although native to our study area, are present today as the result of releases for hunting purposes. Among the mammals, the most detected roadkilled species were the European hedgehog *Erinaceus europaeus* (73 roadkilled individuals), the European badger *Meles meles* (37 roadkilled individuals), the red fox *Vulpes vulpes* (35 roadkilled individuals) and the crested porcupine *Hystrix cristata* (27 roadkilled individuals). For birds, the most detected roadkill were the Eurasian jay *Garrulus glandarius* (13 roadkilled individuals), the common pheasant *Phasianus colchicus* (12 roadkilled individuals) and the European blackbird *Turdus merula* (ten roadkilled individuals). For reptiles, the most detected roadkill were the Western green lizard *Lacerta bilineata* (nine roadkilled individuals), the Aesculapian ratsnake *Zamenis longissimus* (nine roadkilled individuals) and the green whip snake *Hierophis viridiflavus* (eight roadkilled individuals). To conclude, the most detected amphibians were the common toad *Bufo bufo* (31 roadkilled individuals), the agile frog *Rana dalmatina* (nine roadkilled individuals) and the green toad *Bufotes balearicus* (six roadkilled individuals). Roadkill rates for each recorded species are provided in Appendix A.

### 3.2. Literature Review and Analysis

In our first literature review, we retrieved a total of 2314 works from WOS. Among those, only 41 (including our original data) fitted our requirements (i.e., including a total list of roadkilled terrestrial vertebrates, including the explicit numbers of dead individuals for each species) and were included in our analysis. Retained studies included up to 20.00% of introduced species (mean ± SD = 5.30 ± 5.69%). Most retained studies were conducted in tropical and subtropical areas (i.e., 20 studies from 11 countries), with a peak in American countries (i.e., ten studies from three countries; Figure 2). All retained studies included 5434 species (mean ± SD per study = 153.53 ± 432.36) for a total of 276,960 roadkilled individuals (mean ± SD per study = 7177.80 ± 25,451.81). Among those species, up to 17.85% were domestic (mean ± SD = 3.03 ± 4.45%), up to 4.62% were paleo-introduced (mean ± SD = 0.86 ± 1.56%) and up to 12.90% (mean ± SD = 1.24 ± 2.49%) were recent introductions. Roadkill rates for each retained study are provided in Appendix A. As a result of our second literature review, we focused on the countries where the 40 retained studies were carried out, obtaining the percentage of introduced species on the total of existing species for all of them (Appendix A).

Concerning our first hypothesis, the biome was one of the most relevant explanatory variables in both model selections. In fact, the univariate model including this factor obtained plausible support in the roadkill model selection (according to [90]) and was the only supported model in the all-species model selection (Table 1). Overall, in both model selections, the percentage of introduced species in Mediterranean and Temperate biomes was higher than in Tropical and Desert biomes (Figure 3). On the other hand, other variables were also supported in the roadkill model selection, especially the latitude, which was the best-supported model (Table 1; with more introduced species at higher latitudes, roughly corresponding to Mediterranean and Temperate biomes; Appendix A). The number of neighbouring countries obtained plausible support in the roadkill model selection (according to [90]), being the second supported model (Table 1; with more introduced species in island countries than in nations having neighbouring countries).

Concerning the analysis focusing on Mediterranean countries (i.e., Italy, Spain, Portugal, and Tunisia), we retained only seven studies (three for Spain, two for Italy, one for Portugal, and one for Tunisia). No differences were observed in the percentage of introduced species on the total of roadkilled species among most of them, but Tunisia did show higher values (i.e., Figure 4). The country model was more supported than the null one (Appendix A).

## 4. Discussion

In this work, we compared our original data collected on roadkilled terrestrial vertebrates from Central Italy with 40 analogous lists proceeding from published studies from throughout the world. This is, to the best of our knowledge, the first attempt at geographical analysis implementing only roadkill data, and in order to investigate the reliability of this kind of data, we also present an analysis implementing more general data proceeding from biological invasion datasets. We found that the percentage of introduced roadkilled species was higher in both the Mediterranean and Temperate biomes compared to other biomes. This is only a first assessment, implementing relatively simple analyses, but in our opinion, this case study can represent an interesting example of the possible value of roadkill datasets for investigating questions related to research fields other than road ecology, such as biological invasions and biogeography in this case.

Roadkill surveys usually represent a reliable information source regarding the common species that can be found in a given area [91,92,93], although remarkable exceptions have been described for species actively avoiding roads [21,94,95]. Nevertheless, invasive vertebrates are often generalist species able to use roads and roadsides as suitable habitats and to take advantage of the related resources (i.e., roadkilled carcasses, garbage; [96,97]), consequently paying the price of high roadkill rates [15,98,99]. Therefore, it is relatively safe to assume that alien species would be well represented in the roadkill records of a given area [34,100] and, consequently, that such data would be suitable for geographical analyses focused on biological invasions. According to this rationale, our analyses showed that the percentage of introduced species (on both the total of roadkilled species and the total of existing species) is higher in Mediterranean and Temperate biomes than in Tropical and Desert biomes. These findings confirm our prediction that the overall amount of introduced species is higher in the Mediterranean biome, as previously suggested by the available scientific literature [101,102,103]. However, the Temperate biome also suffered high levels of species introduction, especially in Central Europe and North America, both of which regions characterised by suitable climates for most vertebrate species and consequently also suitable for their invasions [73,104,105]. These biomes are more anthropised than others [106,107,108], probably because they are widely distributed in countries with early human presence, mostly in the Mediterranean Basin and, more generally, in Europe. In addition, early anthropization can be a major driver of the presence of established populations of introduced species, especially the paleo-introduced ones, such as the fallow deer *Dama dama* and the crested porcupine *Hystrix cristata* in our fieldwork area (e.g., [109,110]). Both the Mediterranean and Temperate biomes also include some of the countries with suitable latitudes (and consequently climates) for allowing growth and reproduction of introduced terrestrial vertebrates [72,73], such as Italy and Spain, but also USA and UK, with all of them providing published studies to our literature review.

The Mediterranean biome, spanning from coastal and interior portions of the USA, Mexico, Chile, South Africa, Australia and the Mediterranean Basin, is also recognised as one of the most imperilled in terms of biodiversity loss due to its sensitivity to biological invasions, land use and climate change [101,111,112]. In this biome, at least in the most sensitive areas such as national parks, monitoring roadkill should be a priority action, in order to detect conservation issues related to road impacts (see, for example, [113], but also [114,115] for examples in other biomes). Additionally, we showed here that these surveys can also provide valuable information about biological invasions, not only about cost-effective early detection in a given location (see a recent example regarding the American mink *Neovison vison* in Central Italy: [116]), but also to conduct geographical analyses and to estimate the percentage of common alien species in a given area. Government authorities and institutions (e.g., protected areas and environmental agencies) should include roadkill surveys in their long-term monitoring schemes, but citizen science can also be involved in collecting a remarkable amount of data [35,117,118]. Importantly, the use of ad hoc projects on online platforms (e.g., iNaturalist) may allow the collection of data throughout the years and from different sites [119,120,121] and to analyse temporal patterns of invasions. Finally, we would also highlight that less than 2% of roadkill studies focusing on all vertebrates (i.e., 40 on 2314) reported a full list of affected species, making it more difficult than expected to use this literature resource, so we would strongly recommend future roadkill studies to publish their species list (and data), in order to increase their interest and the possibility to be used and cited by other researchers.

Our second prediction was not supported by the specific model selection because there were no differences in the percentage of introduced species on the total of roadkilled species among most considered Mediterranean countries (i.e., Italy, Spain, and Portugal), although Tunisia showed higher values (mainly due to domestic species, possibly free-ranging more than in other Mediterranean countries; [122,123]). Thus, although Italy hosts a high number of alien species among terrestrial vertebrates [66,67,124,125], it does not seem to include a significantly higher number of alien species with respect to other Mediterranean countries included in our analysis. However, these results are probably due to the extremely small sample size (i.e., few studies in few Mediterranean countries), which may prevent achieving exhaustive conclusions, but also because the analysed studies from Italy were carried out in rural areas characterised by a relatively low amount of alien species [79,80,81,115], and are perhaps not completely representative of the whole country.

In addition to the small sample size (for both our fieldwork data and the datasets from literature reviews) and the potentially biased locations of the considered study areas, this work surely entails further limitations. Several factors have been described to drive the number of introduced species in a given country, such as the trade importance, the amount of different climatic niches and the number of potentially competing native species (e.g., [126,127,128]), but most of them were of difficult implementation considering our relatively small sample size. However, although we acknowledge these study limitations, we believe they do not affect the main idea of this work, which is that roadkill datasets can be used beyond the study of road impacts, and in this case, for a first assessment of biological invasions among different countries.

## 5. Conclusions

In this study, we used our fieldwork data collected on roadkilled terrestrial vertebrates from Central Italy and 40 analogous lists proceeding from published studies from all over the world. Our analysis represents the first attempt at biological invasion risk assessment implementing only roadkill data, and, as a result, the percentage of introduced roadkilled species was higher in both the Mediterranean and Temperate biomes compared to other biomes. This is only a first assessment implementing relatively simple analyses, and we acknowledge several study limitations, but, in our opinion, they do not affect the main idea of this study, which is that roadkill datasets can be used for investigating questions related to research fields other than road ecology, such as in this case biological invasions.

## Figures and Tables

**Figure 1 biology-12-00850-f001:**
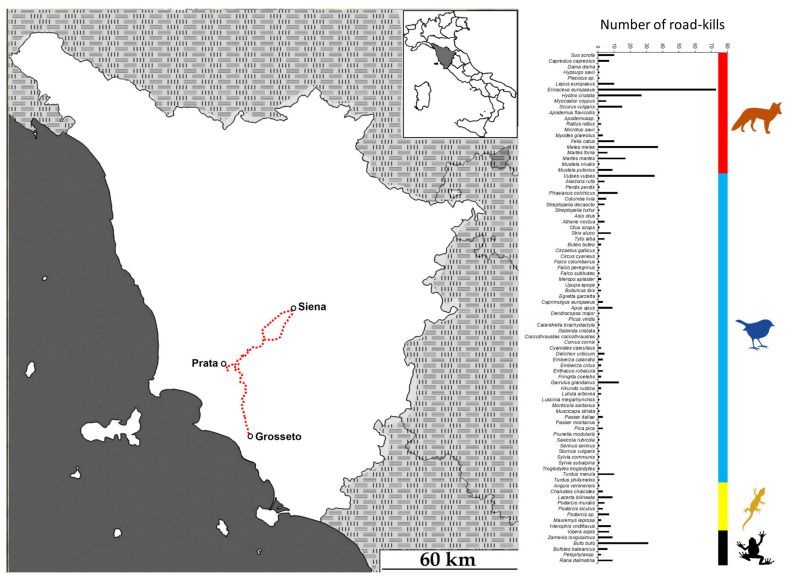
Study area and roadkilled species during the fieldwork survey. On the left is the location of the study area (Central Italy, Tuscany). Roadkill surveys started in the village of Prata (Province of Grosseto) in the direction of Grosseto and then Siena (101 km). On the right is the number of individuals of each roadkilled species.

**Figure 2 biology-12-00850-f002:**
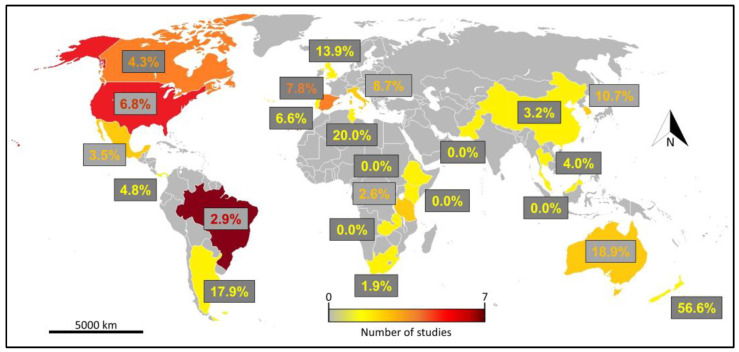
Global literature review of roadkill studies. Country location of retained studies (N = 41). We retained articles focusing on the whole community of terrestrial vertebrates and including an explicit list of roadkilled species. For each considered country, we provided the percentage of introduced species on the total of roadkilled species.

**Figure 3 biology-12-00850-f003:**
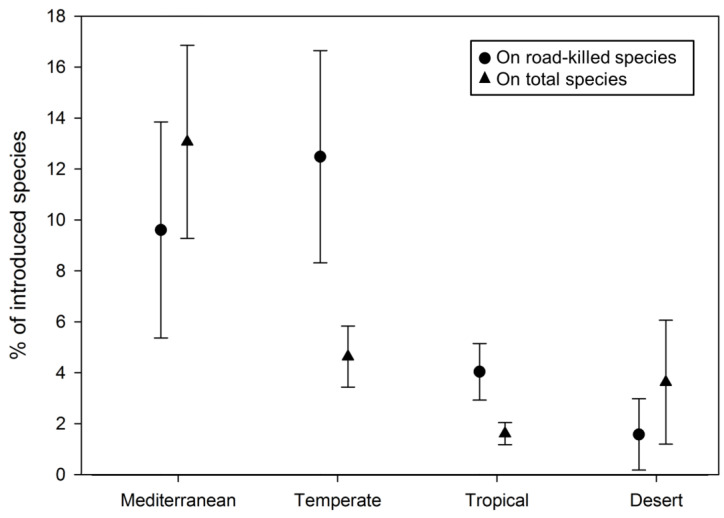
Percentage of introduced species in different biomes, including both the percentage of introduced species on the total of roadkilled species (circles) and on the total of existing species (triangles).

**Figure 4 biology-12-00850-f004:**
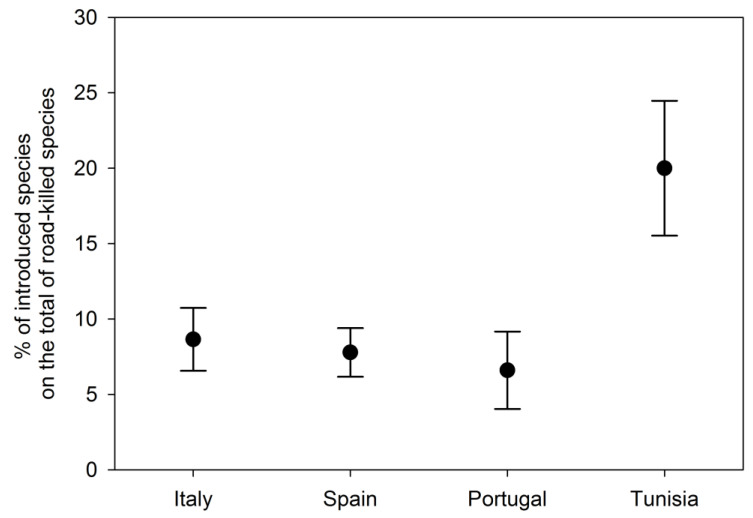
Percentage of introduced species on the total of roadkilled species in different countries, from the model considering all retained studies focusing on Mediterranean biomes.

**Table 1 biology-12-00850-t001:** Factors affecting the percentage of introduced species on the total of roadkilled species (upper table) and the percentage of introduced species on the total of existing species (lower table), model ranks by AICs. The upper table shows the model selection for the roadkill dataset (i.e., literature review on roadkilled terrestrial vertebrates plus our own field data), whereas the lower table shows the model selection for the all-species dataset (i.e., literature review on introduced terrestrial vertebrates in different countries). K is the number of model parameters. ΔAICc is the relative difference between a given AIC value compared to the smallest AICc value. AICc weights (wAICc) indicate the relative support for every model (the weights of all the models in the candidate set have the sum of 1). Evidence ratio (ER) is the ratio of wAIC, comparing the best-supported model with every competing one. Model rank is provided only for the best-supported models. R^2^ is a measure of the global fit of each model. Into the model column, Nc are neighbouring countries.

**Roadkill Model Selection**							
**Model**	**K**	**AICc**	**ΔAICc**	**wAICc**	**ER**	**Rank**	**R^2^**
Latitude	3	244.6	0.0	0.78	1.0	1	0.15
Nc	4	247.9	3.3	0.15	5.0	2	0.21
Biome	5	250.5	5.9	0.04	19.0	3	0.16
Null	2	251.7	7.1	0.02	34.7	-	0.00
GDP	4	256.2	11.6	0.01	324.5	-	0.01
Full	10	258.3	13.6	0.00	918.0	-	0.32
**All-Species Model Selection**							
**Model**		**AICc**	**ΔAICc**	**wAICc**	**ER**	**Rank**	**R^2^**
Biome	5	187.9	0.0	0.99	1.0	1	0.19
Latitude	3	197.3	9.3	0.01	107.0	-	0.54
Full	10	199.2	11.2	0.00	276.7	-	0.57
Nc	4	205.5	17.6	0.00	6555.3	-	0.23
Null	2	210.2	22.3	0.00	70,829.4	-	0.00
GDP	4	214.1	26.1	0.00	475,931.8	-	0.02

## Data Availability

All data generated or analyzed during this study are included in this published article.

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
