# Peer review of "Aliens on the Road: Surveying Wildlife Roadkill to Assess the Risk of Biological Invasion"

_biology, 2023, doi:10.3390/biology12060850_

Round 1
Reviewer 1 Report
Dear authors,
It was a fascinating idea to evaluate introduced fauna through road-kill. I really liked the manuscript. I made some suggestions through the manuscript that is attached, some of them I highlight here.
- Study area: You need to improve figure 1 quality.
- Data collection and analyses: Continuous variables (latitude and the number of neighboring countries) must be standardized before performing any analysis.
Did you perform any post-hoc test to state which group is different?
- Results: Instead of using the number of individuals road-killed I think you should bring the road-kill rate for each species. Even though you did not perform any experiment to correct the road-kill rate, showing road-kill rates would enable future comparisons with other research, as they "correct" the bias of survey effort.
I think you can make more use of the data you have. For example, bring the exact number of introduced species road-killed and the most common ones.
I have a suggestion that will bring some more information to your research (and some extra work too, sorry). You could compare the number of introduced species you identified road-killed with each country's official list of introduced species. This way you will show how reliable is using road-kill data in this field. Supporting even the arguments you used in discussion.
Please, sort models by deltaAICc, this way it is much easier to read the table.
- Discussion: If you accept my suggestions, you will have more information to support your discussion.
One more time, congratulation on your manuscript.
Best of luck.

Author Response
Reviewer1, Query1: Dear authors, it was a fascinating idea to evaluate introduced fauna through road-kill. I really liked the manuscript. I made some suggestions through the manuscript that is attached, some of them I highlight here.
Reply1: Dear Reviewer 1, thank you for your support and useful suggestions. We have modified the manuscript by taking into account all of them. Please find all the modifications in red along the text.
Reviewer1, Query2: Study area: You need to improve figure 1 quality.
Reply2: Figure 1 has been now re-submitted with the highest possible quality.
Reviewer1, Query3: Data collection and analyses: continuous variables (latitude and the number of neighboring countries) must be standardized before performing any analysis.
Reply3: The number of neighboring countries is now an ordinal variable due to convergence issues, and consequently there is no need of standardizing the only remaining continuous variable anymore.
Reviewer1, Query4: Did you perform any post-hoc test to state which group is different?
Reply4: Yes, but we agree with Burnham et al. 2011, so we believe that AIC model selection and p-values should not been implemented in the same manuscript, although this is a widely implemented approach in ecology. Additionally, we believe the results are clear enough to no need post-hoc tests. For all these reasons, we are sorry to say that we would prefer to not include any post-hoc test.
Burnham, K. P., Anderson, D. R., & Huyvaert, K. P. (2011). AIC model selection and multimodel inference in behavioral ecology: some background, observations, and comparisons. Behavioral Ecology and Sociobiology, 65, 23-35.
Reviewer1, Query5: Results: Instead of using the number of individuals road-killed I think you should bring the road-kill rate for each species. Even though you did not perform any experiment to correct the road-kill rate, showing road-kill rates would enable future comparisons with other research, as they "correct" the bias of survey effort.
Reply5: We could not implement road-kill rates in our analyses for convergence issues, but now we included them in Supplementary Material 1.
Reviewer1, Query6: I think you can make more use of the data you have. For example, bring the exact number of introduced species road-killed and the most common ones.
Reply6: We tried to include this information, but it was a huge amount of information, actually confusing the readers from the main narrative, so we finally decided to include only part of it in the Results.
Reviewer1, Query7: I have a suggestion that will bring some more information to your research (and some extra work too, sorry). You could compare the number of introduced species you identified road-killed with each country's official list of introduced species. This way you will show how reliable is using road-kill data in this field. Supporting even the arguments you used in discussion.
Reply7: This information is now included in the manuscript.
Reviewer1, Query8: Please, sort models by deltaAICc, this way it is much easier to read the table.
Reply8: Table 1 has been modified as suggested.
Reviewer1, Query9: Discussion: If you accept my suggestions, you will have more information to support your discussion. One more time, congratulation on your manuscript. Best of luck.
Reply9: All the suggestions have been carefully considered and almost all of them accepted. Thank you again for your support and useful suggestions.
Reviewer 2 Report
In this study, the authors conducted both a field study and literature review to test if the percentage of roadkilled species that are invasive (1) varies with biome and (2) is higher in Italy than in other Mediterranean countries. The goal of the study is to demonstrate that roadkill datasets can be used to examine patterns of biological invasions. Unfortunately, I cannot recommend publication of this paper without major revisions.
First, the study design does not allow the authors to address the paper's goal. To demonstrate the usefulness of roadkill data in examining invasion patterns, the authors would need demonstrate that the percentages of roadkill species that are invasive species in each area are actually representative of percentages of species that are invasive in each area. To do this, the authors could compare the percentage of roadkill species that are invasive to the percentage of species with geographic ranges overlapping the studied road that are invasive. The authors have not done this.
Another problem with this paper is that the authors do not provide a robust reasoning for their second hypothesis. As a result, testing of the second hypothesis seemed completely unnecessary, and the field portion of the study could easily be cut from the paper. The authors also failed to provide robust reasoning for the inclusion of their "confounding variable" as predictors (latitude, number of countries, GDP). In this case, I could guess how each of these predictors might influence the percentage of roadkill species that are invasive, but the information should have been provided early in the paper (not just in the Discussion).
I also noted problems with the statistical analyses. When presenting tables of models that are compared using AICc, the authors absolutely need to include information on the number of parameters estimated by each model. I am assuming the authors did not include interactions between predictors in their models, but that should be explicitly stated. Even without interactions included, the number of parameters estimated by a model including 3 or 4 predictors may be too large for the relative small sample size in this study. This could result in quite a bit of error, and the authors should include SEs or Confidence Intervals to investigate. In their current form, the tables of results indicate analyses that cannot be trusted. The complete dataset reveals none of the models examined provide evidence for support (Delta AICc values are all enormous!). The contemporary dataset reveals only the full model is supported (according to the rule of thumb where Delta AICc is less than 2) and none of the nested tables; this is very strange and seems incorrect. Without parameters estimated, I do not trust these analyses.
Last, extensive editing of the English language and style is required.
Author Response
Reviewer2, Query1: In this study, the authors conducted both a field study and literature review to test if the percentage of roadkilled species that are invasive (1) varies with biome and (2) is higher in Italy than in other Mediterranean countries. The goal of the study is to demonstrate that roadkill datasets can be used to examine patterns of biological invasions. Unfortunately, I cannot recommend publication of this paper without major revisions.
Reply1: All the suggestions made by Reviewer 2 have been addressed in this resubmitted version.
Reviewer2, Query2: First, the study design does not allow the authors to address the paper's goal. To demonstrate the usefulness of roadkill data in examining invasion patterns, the authors would need demonstrate that the percentages of roadkill species that are invasive species in each area are actually representative of percentages of species that are invasive in each area. To do this, the authors could compare the percentage of roadkill species that are invasive to the percentage of species with geographic ranges overlapping the studied road that are invasive. The authors have not done this.
Reply2: The manuscript now includes a comparison with the introduced species list for each considered country.
Reviewer2, Query3: Another problem with this paper is that the authors do not provide a robust reasoning for their second hypothesis. As a result, testing of the second hypothesis seemed completely unnecessary, and the field portion of the study could easily be cut from the paper.
Reply3: We modified the Introduction in order to improve the reasoning supporting our second prediction. We carefully considered if excluding our own road-kill data, which for sure complicated a bit the narrative, but our sample size was so reduced that we finally preferred maintaining our data in the manuscript.
Reviewer2, Query4: The authors also failed to provide robust reasoning for the inclusion of their "confounding variable" as predictors (latitude, number of countries, GDP). In this case, I could guess how each of these predictors might influence the percentage of roadkill species that are invasive, but the information should have been provided early in the paper (not just in the Discussion).
Reply4: We modified both the Introduction and Materials and Methods in order to improve the reasoning supporting the inclusion of confounding variables as predictors.
Reviewer2, Query5: I also noted problems with the statistical analyses. When presenting tables of models that are compared using AICc, the authors absolutely need to include information on the number of parameters estimated by each model. I am assuming the authors did not include interactions between predictors in their models, but that should be explicitly stated. Even without interactions included, the number of parameters estimated by a model including 3 or 4 predictors may be too large for the relative small sample size in this study. This could result in quite a bit of error, and the authors should include SEs or Confidence Intervals to investigate. In their current form, the tables of results indicate analyses that cannot be trusted. The complete dataset reveals none of the models examined provide evidence for support (Delta AICc values are all enormous!). The contemporary dataset reveals only the full model is supported (according to the rule of thumb where Delta AICc is less than 2) and none of the nested tables; this is very strange and seems incorrect. Without parameters estimated, I do not trust these analyses.
Reply5: The number of parameters has been included now. We modified some variables, re-organized the analytical approach, and simplified the model selections.
Reviewer2, Query6: Last, extensive editing of the English language and style is required.
Reply6: A second native English speaker reviewed the English for this resubmitted version.
Reviewer 3 Report
Dear editor and authors
The study entitled “Aliens on the road: surveying wildlife road-kills for assessing the risk of biological invasion” provides new and valuable data about how road-kill datasets can be used to investigate patterns of biological invasions. Despite the extremely small sample size (for both fieldwork data and the dataset from literature review), the authors adequately exploited the data, and promote a consistent discussion. Data are original, and results are convincing and very relevant for publication. In this sense, manuscript has merits and fits completely within the scope of the journal. I didn't find problems or errors on research design or in methods. The results are clearly presents and discussions and conclusions are supported by results. The overall merit of the manuscript is high. For these reasons, I recommend the accept in a present form.
Sincerely,
Reviewer
Author Response
Reviewer 3, Query 1: Dear editor and authors, the study entitled “Aliens on the road: surveying wildlife road-kills for assessing the risk of biological invasion” provides new and valuable data about how road-kill datasets can be used to investigate patterns of biological invasions. Despite the extremely small sample size (for both fieldwork data and the dataset from literature review), the authors adequately exploited the data, and promote a consistent discussion. Data are original, and results are convincing and very relevant for publication. In this sense, manuscript has merits and fits completely within the scope of the journal. I didn't find problems or errors on research design or in methods. The results are clearly presents and discussions and conclusions are supported by results. The overall merit of the manuscript is high. For these reasons, I recommend to accept in the present form. Sincerely, Reviewer.
Reply 1: Dear Reviewer 3, thank you for your review and support.
Round 2
Reviewer 2 Report
The authors have done a very nice job of addressing comments submitted by reviewers, and the manuscript is much improved. Readers will find value in the current version of this manuscript, and I recommend it for publication.